# Presenilin-1-Derived Circular RNAs: Neglected Epigenetic Regulators with Various Functions in Alzheimer’s Disease

**DOI:** 10.3390/biom13091401

**Published:** 2023-09-17

**Authors:** Nima Sanadgol, Javad Amini, Cordian Beyer, Adib Zendedel

**Affiliations:** 1Institute of Neuroanatomy, RWTH University Hospital Aachen, 52074 Aachen, Germany; 2Department of Physiology and Pharmacology, School of Medicine, North Khorasan University of Medical Sciences, Bojnurd 94149-75516, Iran; 3Department of Biomedicine, Institut of Anatomy, University of Basel, 4031 Basel, Switzerland

**Keywords:** Alzheimer’s disease, PSEN1, circular RNAs, miRNA sponging, in silico analysis

## Abstract

The presenilin-1 (PSEN1) gene is crucial in developing Alzheimer’s disease (AD), a progressive neurodegenerative disorder and the most common cause of dementia. Circular RNAs (circRNAs) are non-coding RNA generated through back-splicing, resulting in a covalently closed circular molecule. This study aimed to investigate PSEN1-gene-derived circular RNAs (circPSEN1s) and their potential functions in AD. Our in silico analysis indicated that circPSEN1s (hsa_circ_0008521 and chr14:73614502-73614802) act as sponge molecules for eight specific microRNAs. Surprisingly, two of these miRNAs (has-mir-4668-5p and has-mir-5584-5p) exclusively interact with circPSEN1s rather than mRNA-PSEN1. Furthermore, the analysis of pathways revealed that these two miRNAs predominantly target mRNAs associated with the PI3K-Akt signaling pathway. With sponging these microRNAs, circPSEN1s were found to protect mRNAs commonly targeted by these miRNAs, including QSER1, BACE2, RNF157, PTMA, and GJD3. Furthermore, the miRNAs sequestered by circPSEN1s have a notable preference for targeting the TGF-β and Hippo signaling pathways. We also demonstrated that circPSEN1s potentially interact with FOXA1, ESR1, HNF1B, BRD4, GATA4, EP300, CBX3, PRDM9, and PPARG proteins. These proteins have a prominent preference for targeting the TGF-β and Notch signaling pathways, where EP300 and FOXA1 have the highest number of protein interactions. Molecular docking analysis also confirms the interaction of these hub proteins and Aβ42 with circPSEN1s. Interestingly, circPSEN1s-targeted molecules (miRNAs and proteins) impacted TGF-β, which served as a shared signaling pathway. Finally, the analysis of microarray data unveiled distinct expression patterns of genes influenced by circPSEN1s (WTIP, TGIF, SMAD4, PPP1CB, and BMPR1A) in the brains of AD patients. In summary, our findings suggested that the interaction of circPSEN1s with microRNAs and proteins could affect the fate of specific mRNAs, interrupt the function of unique proteins, and influence cell signaling pathways, generally TGF-β. Further research is necessary to validate these findings and gain a deeper understanding of the precise mechanisms and significance of circPSEN1s in the context of AD.

## 1. Introduction

Alzheimer’s disease (AD) is a neurodegenerative disorder characterized by the loss of brain cells and is the leading cause of dementia, accounting for over 80% of dementia cases [1]. While the mortality rates for heart disease and stroke have decreased in the United States of America, the percentage of deaths attributed to AD has increased by 89% between 2000 and 2014 [2]. AD begins with mild memory problems and progresses to cognitive impairment and daily task difficulties [3], and its prevalence is closely tied to aging [4]. Familial AD (FAD) and sporadic AD, despite some differences, share several fundamental characteristics. Both forms can onset in mid-life or earlier and follow a dominant Mendelian genetic pattern, with close to 100% penetrance in most pedigrees. The three known genes linked to FAD are Presenilin-1 and Presenilin-2 (PSEN1 and PSEN2) and the amyloid precursor protein (APP) [5]. Amyloid β (Aβ), a component of extracellular plaques, particularly Aβ peptide (Aβ-42), is predominant in the medial temporal lobe and cortex of AD brains, and it progressively spreads to deep gray nuclei, the brainstem, and eventually to the cerebellum [6]. In generating Aβ, the transmembrane neuronal protein APP is sequentially cleaved by β-secretase and γ-secretase. PSEN1 and PSEN2 are crucial components of the γ-secretase complex [7]. Despite their similar genomic sequences, structures, and functions, mutations in PSEN1 are more detrimental than mutations in PSEN2 [8,9]. Circular RNAs (circRNAs) are a class of non-coding RNAs that have been discovered in higher eukaryotes. They are generated through “back-splicing”, where an exon’s 3′ splice site is joined to its upstream 5′ splice site on the same RNA molecule [10,11]. While circRNAs are generally considered a class of non-coding RNAs, there have been instances where certain circRNAs can form circular open reading frames (ORFs) via backspacing. These circular ORFs can potentially be translated into proteins. However, such instances are relatively rare compared to the broader circRNAs, and most circRNAs are indeed non-coding [12]. CircRNAs can be categorized into three types based on their splicing patterns and the presence of exons/introns: circular intronic RNAs, exon–intron circRNAs, and exonic circRNAs [13]. Due to their covalently closed loop structure, circRNAs resist exonuclease-mediated degradation, resulting in a longer half-life than linear mRNAs [14]. CircRNAs interact with non-coding RNAs and play various roles, such as serving as microRNA sponges, protein scaffolds, splicing competitors, and triggers for protein nuclear translocation [15]. A recent study investigated cortex samples from postmortem AD patients and identified several PSEN1-gene-derived circular RNAs (circPSEN1s). Among them, two circPSEN1s (hsa_circ_0008521 and hsa_circ_0003848) were recently found to be upregulated in the brain cortex of AD despite no significant change in PSEN1 gene expression compared to controls [16]. The present study aims to predict the functions of these two circRNAs with presumably important roles in AD.

## 2. Materials and Methods

The information regarding circPSEN1s (hsa_circ_0008521 and hsa_circ_0003848), including their identification and sequences, was obtained from two different databases. The circBase database was utilized to identify known and novel circRNAs in sequencing data. It provides access to comprehensive datasets of circRNAs along with supporting evidence of their expression within the genomic context [17]. The sequences of hsa_circ_0008521 and hsa_circ_0003848 were specifically extracted from the UCSC database [18].

### 2.1. circPSEN1s–miRNAs and circRNA–Protein Interaction Prediction

To predict the interactions between circPSEN1s and miRNAs, the sequences of the circRNAs were uploaded to circAtlas, a comprehensive database for circRNA analysis. The circAtlas uses three prediction tools, targetScan, miRanda, and Pita, to predict circRNA–miRNA interactions. For this study, common miRNAs were selected that were predicted by at least two of these prediction tools. Additionally, circAtlas provides the capability to predict circRNA–protein interactions. This prediction is based on two models: one utilizes RNA flanking information, and the other is based on the RNA sequence itself. This study combined both models to predict circRNA–protein interactions [19]. HMDD 3.2, the human microRNA disease database, validated the relationship between the identified miRNAs and the disease. HMDD is a curated resource that provides information on the association between miRNAs and diseases [20]. By utilizing HMDD 3.2, we verified the relationship between the predicted miRNAs and the disease of interest, likely AD in this case.

### 2.2. miRNA Gene-Target Prediction and Pathway Enrichment Analysis

We utilized the Mienturnet database to predict the miRNAs’ gene targets. Mienturnet is a database that provides information on miRNA–target interactions. A threshold of *p*-value < 0.05 was considered to select significant miRNA gene-target interactions [21]. The miRNA gene targets and proteins that interact with circPSEN1s were subjected to pathway analysis for the analysis of cellular pathway enrichment. Enrichr, a web-based tool for pathway enrichment analysis, was used to identify potential KEGG (Kyoto Encyclopedia of Genes and Genomes) cellular pathways and gene ontology associated with the miRNA gene targets and proteins. A significance threshold of *p*-value < 0.05 was applied to determine the enriched pathways based on the Enrichr results [22]. To investigate the relationship between the identified genes and human disorders, including AD, dementia, and neurodegenerative disorders, we referred to the DisGeNET database (https://www.disgenet.org/, accessed on 15 March 2023). DisGeNET is a comprehensive collection of genes and their associations with various human disorders. By utilizing the database, we could identify genes linked to AD and other related disorders [23].

### 2.3. Protein–Protein Interaction (PPI)

To explore protein–protein interactions (PPIs), we constructed a PPI network using the STRING database version 12.0. STRING is a comprehensive database integrating known and predicted protein–protein interactions from various sources. We obtained information on the potential interactions between these proteins by inputting the miRNA gene targets and proteins that may interact with circPSEN1s into the STRING database [23]. To analyze and visualize the results of the STRING PPI network, we utilized Cytoscape v3.8.2. Cytoscape is a powerful software platform for visualizing and analyzing complex networks, including biological networks like PPI. It allows for the visualization of nodes (proteins) and edges (interactions) in the network, as well as the application of various analytical methods to gain insights into the network structure and function [23]. By combining the information from STRING and the analytical capabilities of Cytoscape, we explored and analyzed the protein–protein interactions associated with the miRNA gene targets and proteins interacting with circPSEN1s visually and interactively.

### 2.4. circPSEN1s and Protein Molecular Docking

The secondary structure of circPSEN1s was determined using the RNAfold server, and the results were downloaded in Vienna format [24]. The secondary structure information is crucial for predicting the three-dimensional (3D) structure of circRNAs. For this purpose, we employed the 3DRNA web service, which utilizes both the sequences and secondary structures of circRNAs to predict their 3D structures [25]. Among the proteins that were found to interact with circPSEN1s directly, hub proteins were selected for molecular docking. The co-crystal structures of these hub proteins and Aβ42 peptide (as a key pathological feature of AD present frequently in amyloid plaques) were obtained from the Protein Data Bank (PDB), a comprehensive repository of experimentally determined protein structures (https://www.rcsb.org/, accessed on 15 March 2023) [26]. Specifically, we downloaded the PDB IDs 3BIY, 7VOX, and 1IYT, corresponding to EP300 (E1A binding protein p300), FOXA1 (forkhead box A1), and Aβ42, respectively. To prepare the crystal structures for molecular docking, we used Chimaera 1.8.1, a molecular modeling system, to clean up the structures by removing ligands, ions, and solvents [27]. Finally, the molecular docking process was carried out using the HDOCK server [28]. HDOCK is an online platform that performs protein–protein docking to predict the possible interactions between proteins of interest. By employing these computational tools and databases, we aimed to gain insights into the 3D structure of circPSEN1s and predict the potential interactions between circPSEN1s and hub proteins (chosen based on previous PPI analysis), including EP300 and FOXA1, as well as Aβ42.

### 2.5. Microarray Data Analysis of Postmortem AD Patients

To investigate the expression patterns of genes related to AD, we accessed the Gene Expression Omnibus (GEO) database provided by the National Center for Biotechnology Information (NCBI) [29]. Specifically, we extracted the expression profile dataset GSE48350 from GEO, which contains gene expression information from postmortem AD patients. This dataset consists of 253 samples obtained from four different brain areas: entorhinal cortex (*n* = 54), hippocampus (*n* = 62), postcentral gyrus (*n* = 68), and superior frontal gyrus (*n* = 69). To analyze the gene expression data, we utilized the online program GEO2R, which is available through the GEO database website (https://www.ncbi.nlm.nih.gov/geo/geo2r/, accessed on 15 March 2023) [30]. GEO2R is based on the “limma” R package, a widely used tool for analyzing differential gene expression. By using GEO2R, we were able to identify genes with differential expression in the GSE48350 microarray data. The Benjamini–Hochberg procedure was employed to adjust *p*-values and manage the false discovery rate. A *p*-value threshold of 0.05 was applied without implementing a fold change threshold. An analysis of variance (ANOVA) was utilized to compare more than two groups and ascertain variations in expression levels. It is important to note that all samples in the GSE48350 dataset were included in the analysis to evaluate the expression of genes that may be associated with or influenced by circPSEN1s. By examining gene expression profiles in human AD samples, we aimed to understand better the potential interactions and regulatory mechanisms involving circPSEN1s and the genes with differential expression. We sought to validate our prior in silico findings.

## 3. Results

The circRNA hsa_circ_0008521 is located in the genomic region chr14:73614502-73614802 and has a length of 210 base pairs (bp). On the other hand, hsa_circ_0003848 spans the region chr14:73614502-73614814 and has a length of 222 bp (Figure 1A). According to circBase, both hsa_circ_0008521 and hsa_circ_0003848 are expressed in various brain regions, including the cortex, frontal cortex, parietal lobe, temporal lobe, cerebellum, and SY5Y cells. This indicates that these circRNAs have a widespread expression pattern within the brain and neuronal cell lines. Investigating whether these circRNAs are exclusive to the brain is crucial. Additionally, it is pertinent to discern whether these circRNAs are primarily abundant in neurons or if they also exist within glial cells. This insight could shed light on their potential roles and functions in AD.

### 3.1. circPSEN1s–miRNAs Interaction Prediction

The miRNAs that were consistently identified across various databases (Figure 1B) to interact with both hsa_circ_0008521 and hsa_circ_0003848 included hsa-miR-597-3p, hsa-miR-616-3p, has-miR-1972, has-miR-4279, hsa-miR-4668-5p, hsa-miR-4687-3p, hsa-miR-5584-5p, and hsa-miR-6755-3p (Figure 1C). The circular RNA’s sequence is relatively similar to the gene’s mRNA, except at the junction site, where the two ends of the circular RNA provide a new sequence. Based on this, we initially predicted that all eight miRNAs would interact with both the circular RNA and the mRNA of the gene. However, it is surprising that two of these miRNAs (has-mir-4668-5p and has-mir-5584-5p) exclusively interact with circPSEN1s, not mRNA-PSEN1 (Figure 1C). Additionally, pathway analysis indicated that these two miRNAs primarily target mRNAs linked to the PI3K-Akt signaling pathway (Figure 1D). It is worth noting that no specific diseases were associated with these miRNAs based on information in the HMDD v3.2 database. This suggests that their roles and implications in disease processes, including AD, may have yet to be extensively studied or reported in the available literature within the database.

#### 3.1.1. miRNA Gene-Target Prediction and Pathway Enrichment Analysis

Out of the 237 significant gene targets identified for the eight miRNAs, 27 were associated with AD, according to the DisGeNET database. Furthermore, 13 genes were related to dementia, and 9 were associated with neurodegenerative disorders (Figure 2A). Among all the gene targets, QSER1 (glutamine and serine-rich 1), BACE2 (beta-secretase 2), RNF157 (ring finger protein 157), PTMA (prothymosin alpha), and GJD3 (gap junction protein delta 3) showed the highest number of interactions with the miRNAs (Figure 2B). These genes will likely play important roles in the regulatory network mediated by the identified miRNAs and may have implications in AD and other related disorders.

#### 3.1.2. Pathway Enrichment Analysis and PPI Network Analysis of Proteins

Among the cellular pathways analyzed, three pathways were identified as being associated with the gene targets of the miRNAs. These pathways are the cell cycle, the Hippo signaling pathway, and the TGF-β (transforming growth factor β) signaling pathway (Figure 3A). Interestingly, the Hippo signaling pathway was found to be the most common target for the miRNAs that are sponged by circPSEN1s. This suggests that the interaction between circPSEN1s and the miRNAs may significantly impact the regulation of the Hippo signaling function. As illustrated, SMAD4 (a member of the SMAD family, specifically SMAD family member 4) is the sole gene shared across all three signaling pathways (Figure 3A). Furthermore, a PPI network was constructed using the STRING database and analyzed using Cytoscape (Figure 3B). In this network, three proteins, PPP1CB (protein phosphatase 1 catalytic subunit beta), INCENP (inner centromere protein), and MRPS15 (mitochondrial ribosomal protein S15), exhibited the highest protein interactions. These interactions highlight their potential involvement in the indirect regulatory mechanisms associated with circPSEN1s. It could also indicate that the specific set of miRNAs and gene targets analyzed in this study do not exhibit extensive PPIs or may be involved in different regulatory mechanisms. Further exploration and analysis using alternative methods or datasets could provide more insights into these gene targets’ functional relationships and interactions. Gene ontology analysis investigated the functional characteristics of miRNAs interacting with circPSEN1s (Figure 3C). Further investigation into the functional implications of this interaction in the context of AD and other related conditions would be recommended. Figure 4 summarizes the predicted function of miRNAs interacting with circPSEN1s based on our in silico analysis.

### 3.2. Potential circPSEN1 Protein Targets

The proteins that directly interacted with both hsa_circ_0008521 and hsa_circ_0003848 were investigated using two models: the flank model and the sequence model. Twenty-three proteins were identified for the flank model, while the sequence model yielded fifty-five proteins. Among these, nine proteins were common between the two models. These proteins are FOXA1, estrogen receptor 1 (ESR1), HNF1 homeobox B (HNF1B), bromodomain containing 4 (BRD4), GATA binding protein 4 (GATA4), EP300, chromo box 3 (CBX3), PR/SET domain 9 (PRDM9), and peroxisome proliferator-activated receptor gamma (PPARG). These proteins play important roles in various biological processes and may have potential direct interactions with the circRNAs hsa_circ_0008521 and hsa_circ_0003848.

#### 3.2.1. Pathway Enrichment Analysis and PPI Network Analysis of Proteins

Pathway enrichment analysis was conducted for proteins directly interacting with circPSEN1s, identifying six significant cellular pathways. These pathways include the adherens junction, long-term potentiation, longevity regulating pathway, lysine degradation, Notch signaling pathway, and TGF-β signaling pathway (Figure 5A). The targeting preference of proteins directly interacting with circPSEN1s prominently lies in the Notch signaling pathway. As illustrated, EP300 is the exclusive gene identified within four distinct signaling pathways (Figure 5A). Furthermore, a PPI network was constructed using the STRING database and analyzed using Cytoscape (Figure 5B). Within this network, two proteins, EP300 and FOXA1, exhibited the highest number of protein interactions. These interactions highlight the potential involvement of EP300 and FOXA1 in the regulatory mechanisms associated with circPSEN1s. Gene ontology analysis was performed to investigate the functional characteristics of proteins interacting with circPSEN1s (Figure 5C). Further investigation and analysis of these pathways and protein interactions can provide valuable insights into the functional roles and molecular mechanisms of circPSEN1s and their potential implications in biological processes and diseases, such as AD.

#### 3.2.2. Molecular Docking

The molecular docking analysis was performed to assess the interaction of circPSEN1s (hsa_circ_0008521 and hsa_circ_0003848) with hub proteins (EP300 and FOXA1) and Aβ42. As mentioned, EP300 remains the sole gene identified within four distinct signaling pathways. An added benefit of analyzing Aβ42 lies in its potential to supplement previously demonstrated circPSEN1-hub protein interactions. Unlike databases that utilized the APP sequence for interaction prediction, Aβ42 was derived from the APP and is notably predisposed to aggregation, frequently associated with AD pathology. Consequently, this analysis offers insights into the plausible interaction between circPSEN1s and these peptides. The results showed that both circRNAs could bind to these three proteins (Figure 6). However, the interaction scores hsa_circ_0008521 with FOXA1, EP300, and Aβ42 were higher than hsa_circ_0003848. These docking scores indicate the strength of the interactions between the circRNAs and the proteins. The higher negative scores for hsa_circ_0008521 suggest a potentially stronger binding affinity than hsa_circ_0003848. These findings provide insights into the potential molecular interactions and binding capabilities of circPSEN1s with hub proteins and Aβ42, which may affect their functional roles and involvement in disease processes. 

### 3.3. circPSEN1-Related Genes in the Brain of AD Patients

The microarray data analysis revealed distinct gene expression patterns influenced by circPSEN1 (hsa_circ_0008521) in various brain regions (Figure 7). In the entorhinal cortex, PPP1CB showed significant downregulation, while SMAD4 and WTIP (WT1 interacting protein) displayed altered expression without the specific direction mentioned. On the other hand, BMPR1A (bone morphogenetic protein receptor type 1A) was significantly upregulated, and TGIF1 (TGFB-induced factor homeobox 1) exhibited variable expression changes. SMAD4 and BMPR1A were significantly upregulated in the hippocampus, and TGIF1 showed increased expression. The postcentral gyrus exhibited downregulation of PPP1CB (protein phosphatase 1 catalytic subunit beta) and upregulation of TGIF1. In the superior frontal gyrus, PPP1CB was downregulated with a significant *p*-value, and TGIF1 showed increased expression. These findings highlight the region-specific expression alterations of these genes, indicating their potential involvement in the pathogenesis of AD and related disorders.

Figure 8 provides a comprehensive overview of the anticipated functions of proteins that interact with circPSEN1s, as inferred from our in silico analysis. Given the notably substantial *p*-value associated with the Notch pathway, as well as the previously established significance of TGF-β as a pathway indirectly regulated by circPSEN1s’ targeted miRNAs, and considering the comparatively limited relevance of other pathways to AD, we place particular emphasis on depicting these two pathways within the image.

## 4. Discussion

AD is a prevalent neurodegenerative disease with no effective treatment [31,32]. Moreover, mutations in the PSEN1 gene contribute to the aggregation of Aβ42 in the brains of AD patients [33]. Following an extensive literature review and a thorough assessment of circular RNA-seq data obtained from human AD samples, we found that only two studies have reported the upregulation of PSEN1-derived circRNAs in individuals diagnosed with AD [16,34]. In the leading project, the discriminatory potential of circPSEN1 counts was assessed in differentiating individuals with autosomal dominant AD (ADAD) from those with sporadic AD and healthy controls. This indicates that circPSEN1s (hsa_circ_0008521 and hsa_circ_0003848) counts can effectively distinguish between ADAD cases, sporadic AD cases, and healthy individuals, suggesting its potential as a biomarker for AD subtypes [16]. Moreover, Puri et al. (using analysis of two public AD brain RNA-seq databases) demonstrated that circPSEN1 (hsa_circ_0003848) exhibits upregulation in AD and notably correlates with plaque formation. An intriguing aspect of their findings is that the expression of circPSEN1 seems confined to the cortex. Furthermore, it was observed that circPSEN1 exhibited a greater level of responsiveness to oligomeric tau (S1p fraction) compared to linear PSEN1 mRNA. They also provided evidence that the elevated levels of circPSEN1 in AD could potentially lead to a decrease in the abundance of miR-137. This reduction in miR-137 availability could result in the increased expression of target genes. However, our in silico analysis did not confirm this observation [34]. Our study reported that out of the eight miRNAs potentially interacting with circPSEN1, mir-4668-5p and mir-5584-5p are the only ones that exhibit exclusivity to circPSEN1 and do not engage with PSEN1 mRNA. Notably, the sequence responsible for mediating this interaction is present in circPSEN1 and PSEN1 mRNA.

The fact that a miRNA can interact with a circRNA but not with the mRNA of the same gene with the same sequence can be attributed to several aspects. These interactions are complex processes influenced by factors beyond just sequence complementarity. The secondary structure, competition for binding, contextual regulation, and post-transcriptional modifications all contribute to the observed differences in miRNA interactions between these RNA species. Furthermore, our pathway analysis revealed that the mRNAs implicated in the PI3K-Akt signaling pathway are predominantly targeted by each of these two miRNAs, along with their shared target genes. Dysregulation of the PI3K-Akt pathway can increase neuronal vulnerability and cell death in affected brain regions. The PI3K-Akt pathway is involved in the regulation of enzymes that process the amyloid precursor protein (APP). Dysregulation of this pathway can impact the production and clearance of Aβ, contributing to plaque formation. The PI3K-Akt pathway can modulate the activity of kinases and phosphatases that regulate tau phosphorylation, thus influencing its aggregation and impact on neuronal function [35]. Given the multifaceted nature of AD, the PI3K-Akt signaling pathway’s involvement in various cellular processes makes it a crucial target for understanding the disease mechanisms and developing potential therapeutic interventions.

Our analysis found that QSER1, BACE1, RNF157, PTMA, and GJD3 exhibited the highest interactions with the miRNAs targeted by circPSEN1s. The QSER1 gene, or Glutamine and Serine-Rich Protein 1, is a human gene located on chromosome 2q31.1. It encodes a protein that plays a role in various cellular processes, including transcriptional regulation and RNA processing. Knocking down QSER1 has been shown to induce apoptosis in both p53 wild-type and mutant cancer cells. While QSER1’s exact functions and mechanisms of action are still being investigated, it appears to be a significant player in cellular homeostasis and cancer biology [36]. Interestingly, it is worth noting that both mir-4668-5p and mir-5584-5p exhibit interactions with the mRNA of QSER1.

The BACE1 gene, also known as Beta-Site Amyloid Precursor Protein Cleaving Enzyme 1, is a human gene located on chromosome 11q23.2. It encodes an enzyme called beta-secretase 1, which plays a crucial role in processing the APP in the brain. BACE1 cleaves the APP at the beta-secretase site, leading to the generation of beta-amyloid peptides, including the toxic Aβ42 peptide that accumulates in the brains of individuals with AD. Studies have shown that genetic deletion or inhibition of BACE1 in animal models can significantly reduce beta-amyloid levels and ameliorate AD-related cognitive impairments. The interaction between PSEN1 and BACE1 affects the production of different Aβ isoforms, influencing their aggregation propensity. The imbalance between Aβ40 and Aβ42, influenced by the PSEN1-BACE1 interaction, contributes to the toxicity of Aβ aggregates and their impact on neuronal function and viability [37].

The RNF157 gene, or Ring Finger Protein 157, is a human gene located on chromosome 1p35.2. It encodes a protein that belongs to the RBR (Ring Between Ring fingers) family of E3 ubiquitin ligases. E3 ubiquitin ligases play a crucial role in protein ubiquitination, which regulates protein degradation and other cellular processes. RNF157 is involved in various cellular functions and signaling pathways. It has been shown to regulate cell survival and apoptosis, and its ligase activity is essential for these functions. RNF157 acts as a downstream effector of the PI3K (Phosphatidylinositol 3-kinase) and MAPK (Mitogen-Activated Protein Kinase) pathways, which are important signaling pathways involved in cell growth, proliferation, and survival [38,39].

The PTMA gene, or Prothymosin alpha, is a human gene located on chromosome Xq28. It encodes a protein called Prothymosin alpha, which is involved in various cellular processes and has multiple functions within the cell. Prothymosin alpha has been found to interact with High Mobility Group Box 1 (HMGB1), a DNA-binding protein involved in gene regulation, DNA repair, and inflammation [40]. This interaction is believed to be important for controlling mitochondrial oxidative phosphorylation, which is crucial for cellular energy production. In the context of AD, studies have shown that the expression of PTMA is upregulated in multiple brain regions of AD patients, including the entorhinal cortex, hippocampus, middle temporal gyrus, posterior cingulate cortex, and superior frontal gyrus [41]. It is interesting to note that mir-4668-5p and mir-5584-5p do not interact with the PTMA mRNA. The concept of Alzheimer’s disease (AD) being referred to as “type 3 diabetes” due to its connection with insulin resistance and metabolic dysfunction has gained attention in recent years. The interplay between PTMA, mitochondrial function, and AD-related pathways will likely involve multiple molecular mechanisms. Understanding whether PTMA has a net protective or aggravating role would require detailed studies on its precise interactions with mitochondrial components, its downstream effects on cellular processes, and its correlation with disease progression.

The GJD3 gene, or Gap Junction Delta-3 protein or Connexin 30.2, is a human gene located on chromosome 5q35.2. It encodes a protein that belongs to the connexin family of gap junction proteins. It has been implicated in wound healing, where it plays a role in the migration and proliferation of cells involved in tissue repair. Additionally, variations in the GJD3 gene have been associated with developing varicose veins, characterized by enlarged and twisted veins [42]. The exact mechanisms and functions of GJD3 in different tissues and diseases are still under investigation.

Moreover, our in silico functional analyses have revealed that circPSEN1s are implicated indirectly in various biological pathways, including the cell cycle, the Hippo signaling pathway, and the TGF-β signaling pathway via sponging miRNAs. Over the past decade, extensive research has highlighted the significance of cell cycle aberrations as a neuropathological aspect of AD. These anomalies appear early in the disease process, preceding the formation of plaques and tangles. Mounting evidence suggests that the pathophysiology of AD is significantly influenced by the failure of neuronal cell cycle regulation, leading to apoptosis [43]. Moreover, one evolutionarily conserved mechanism involved in regulating apoptosis, known as the Hippo signaling system, has garnered attention for its tumor-inhibiting properties and accelerating neurodegeneration [44]. Studies indicate that disruption of PP2AC (catalytic subunit of protein phosphatase 2A), a crucial member of the protein phosphatase family that negatively regulates the Hippo pathway, contributes to AD-like symptoms [45]. Interestingly, in our study, the Hippo signaling pathway emerged as a particularly intriguing target for the miRNAs sequestered by circPSEN1s among the analyzed cellular pathways. These circRNAs exhibited a strong affinity (lower *p*-value) for the miRNAs associated with regulating the Hippo pathway. Recent findings suggest that the Hippo pathway has the potential to serve as a mechanosensing pathway in microglia and may represent a promising therapeutic target for preventing microglial-induced neurodegeneration in AD [46].

Impaired TGF-1 signaling, associated with increased Aβ deposition and the formation of neurofibrillary tangles, has been implicated in accelerating neurodegeneration. The canonical TGF-1/Smad signal decreases with aging and chronic inflammation, leading to cytotoxic microglia activation and neurodegeneration [47]. In transgenic mice, long-term overexpression of TGF-β by astrocytes promotes the clearance of Aβ plaque by activated microglia and improves Aβ-induced behavioral impairment. TGF-β may also encourage astrocyte aggregation around brain microvessels and deposits on vascular basement membranes. As a result, TGF-β can decrease brain parenchymal and type Aβ pathogenesis while impairing blood flow to nearby areas [48]. We observe that SMAD4 is a gene that appears within all three signaling pathways (cell cycle, Hippo, and the TGF-β). In AD, dysregulation of the immune system and chronic neuroinflammation are believed to contribute to disease progression. SMAD4’s role in the TGF-β pathway could potentially impact these processes. However, it is important to note that the exact mechanisms of SMAD4’s involvement in dementia, including any direct or specific functions it might have, are complex and still being studied.

On the other hand, our study also revealed that circPSEN1s have the potential to directly interact with several proteins, including FOXA1, ESR1, HNF1B, BRD4, GATA4, EP300, CBX3, PRDM9, and PPARG. These proteins demonstrate a notable preference for targeting the six distinct signaling pathways, with Notch exhibiting a lower *p*-value and greater significance among these pathways. This finding suggests that circPSEN1s may play a role in modulating the activity of the Notch signaling pathway through their interactions with these proteins. Neurovascular impairment may be brought on by age-related alterations in Notch signaling and contribute to the emergence of neurodegenerative disorders [49]. Fascinatingly, the TGF-β signaling pathway emerged as a noteworthy shared target for the miRNAs and proteins sequestered by circPSEN1s. We also note that EP300 is a gene appearing in four of these six signaling pathways.

Furthermore, the PPI network analysis revealed that EP300 and FOXA1 exhibit the highest protein interactions. This suggests that these proteins play a central role in the network and are potentially involved in various biological processes. Furthermore, the interaction between these hub proteins and Aβ42 with circPSEN1s was confirmed by molecular docking analysis, providing additional evidence for their functional association. These findings support the hypothesis that circPSEN1s may regulate the activity and function of EP300, FOXA1, and Aβ42 through direct protein interactions.

EP300 is a transcriptional coactivator and a histone acetyltransferase, which plays a crucial role in regulating gene expression by modifying chromatin structure [50]. There appears to be a recurring pattern of significant differences in EP300 amplitude between individuals with probable AD and those in the healthy control group [51]. It has been identified that alterations in EP300 parameters, such as mutations, can lead to the onset of familial AD approximately 10 years earlier than usual [52].

FOXA1, also known as hepatocyte nuclear factor 3-alpha (HNF3A), encodes a transcription factor belonging to the forkhead box (FOX) family, crucial for regulating androgen receptor and steroid receptor functions [53]. A study on the rat model of AD showed that inhibition expression of FOXA1 improves the cognition-damaging impact of sevoflurane on AD [54]. FOXA1 is expressed in the brain, including regions affected by AD pathology. It has been proposed that FOXA1 may modulate oxidative stress pathways and affect the susceptibility of neurons to oxidative damage. Dysregulation of FOXA1 expression or activity may disrupt normal cellular responses to oxidative stress, leading to increased vulnerability of neurons to damage and degeneration in AD [55].

Finally, the analysis of microarray data revealed unique expression patterns of several genes (WTIP, TGIF, SMAD4, PPP1CB, and BMPR1A) and proteins (FOXA1, ESR1, HNF1B, BRD4, GATA4, EP300, CBX3, PRDM9, PPARG) that are influenced by circPSEN1s in the brains of individuals with AD. These genes demonstrate altered expression levels, suggesting their involvement in the pathological processes associated with the disease. Among the four analyzed brain regions (hippocampus, entorhinal cortex, postcentral gyrus, and superior frontal gyrus), the entorhinal cortex showed the highest number of significant changes in the expression of circPSEN1 indirectly related genes (all genes are predicted with our analysis) and direct proteins (five proteins are predicted with our analysis). While a direct correlation between microarray data analysis and circPSEN1 expression might not be feasible, these data indirectly support our hypothesis. Specifically, the observed upregulation of circPSEN1 suggests its potential to sequester certain miRNAs that would otherwise inhibit the translation of WTIP, TGIF, SMAD4, PPP1CB, and BMPR1A genes. Intriguingly, the overexpression of all these genes in the entorhinal cortex aligns with the prediction of increased circPSEN1 expression in this specific tissue. As a prospective avenue of research, we strongly advocate for assessing circPSEN1 in the entorhinal cortex in future studies. Addressing the substantial overexpression of ESR1, HNF1B, BRD4, and GATA4 in the entorhinal cortex and its potential relationship with circPSEN1 expression presents a challenge. It is difficult to ascertain whether the direct interaction of circPSEN1 with these proteins would yield positive or negative effects on their functionality. Further investigations are necessary for future studies to validate the concurrence of their elevated expression with circPSEN1.

Altogether, our data suggest circPSEN1 may have a particularly pronounced impact on gene expression and protein regulation in the entorhinal cortex compared to the other brain regions examined. The entorhinal cortex is a brain region located in the medial temporal lobe. It plays a crucial role in memory and navigation processes. The entorhinal cortex connects the hippocampus, vital for memory formation, with other brain regions. It serves as a gateway for information flowing between the neocortex and the hippocampus, facilitating the transfer of sensory and spatial information. The entorhinal cortex is also implicated in the encoding and retrieval of episodic memories and spatial navigation, making it essential for learning and memory functions. Both preclinical and clinical studies have indicated that the upstream hippocampal circuitry is crucial in connecting the entorhinal cortex with adult neurogenesis. This suggests that the mechanism of neural circuitry and neurogenesis can be a viable concept to explore when addressing deficiencies in memory, pattern separation, and emotional processing, all commonly impaired in individuals with depression [56].

## 5. Conclusions

Indeed, comprehending the specific roles and interactions of circPSEN1s with the reported genes and proteins holds significant potential for gaining valuable insights into the molecular mechanisms involved in Alzheimer’s pathology. This understanding can shed light on the underlying processes contributing to the progression of the disease. It may also pave the way for developing novel therapeutic approaches that target circPSEN1s and their associated pathways. However, further investigations are necessary to delve deeper into the functional implications of circPSEN1s in gene transcription and protein function. Unraveling the precise mechanisms by which circPSEN1s influence these processes will provide a more comprehensive understanding of their significance in AD progression. Such studies can uncover new targets for intervention and offer promising avenues for therapeutic strategies to combat AD.

## Figures and Tables

**Figure 1 biomolecules-13-01401-f001:**
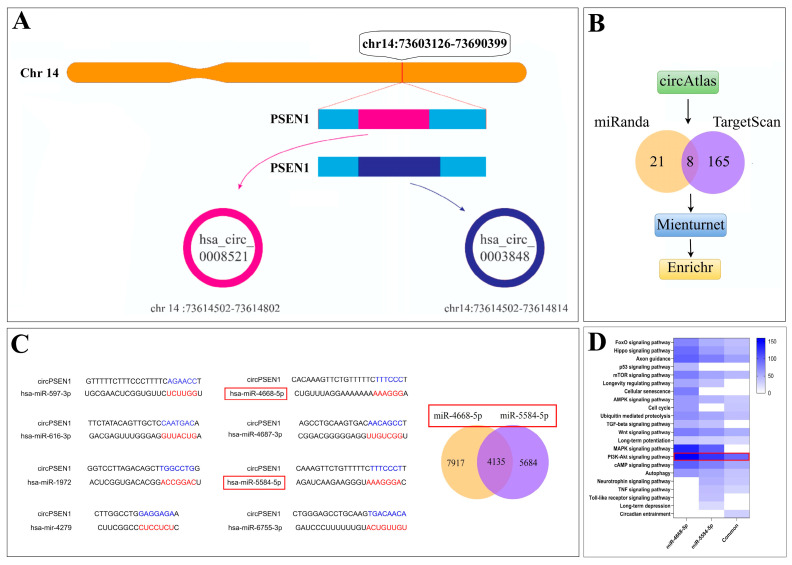
The circRNAs hsa_circ_0008521 and hsa_circ_0003848 are located on chromosome 14, specifically in the genomic regions chr14:73614502-73614802 and chr14:73614502-73614814 of the presenilin-1 (PSEN1) gene respectively (**A**). Eight miRNAs were identified among databases commonly interacting with these circRNAs including hsa-miR-597-3p, hsa-miR-616-3p, has-miR-1972, has-miR-4279, hsa-miR-4668-5p, hsa-miR-4687-3p, hsa-miR-5584-5p, and hsa-miR-6755-3p (**B**,**C**). Further analysis revealed that two of these miRNAs, has-mir-4668-5p and has-mir-5584-5p (indicated with a red rectangle), exclusively interact with circPSEN1s and do not interact with mRNA-PSEN1 ((**C**) left). These two miRNAs mutually interact with 4135 mRNAs ((**C**) right). Moreover, pathway analysis showed that mRNAs involved in the PI3K-Akt signaling pathway are mainly targeted with each of miRNAs and their common targeted genes (**D**).

**Figure 2 biomolecules-13-01401-f002:**
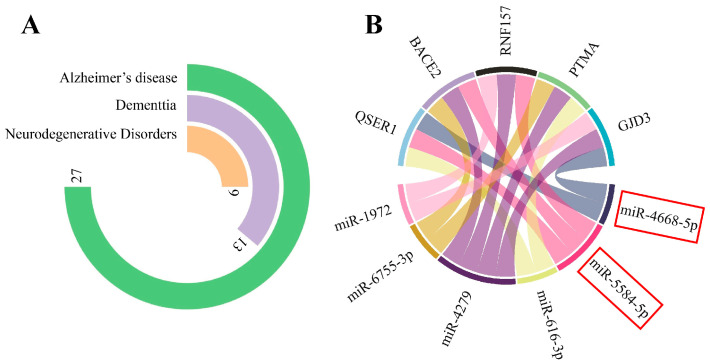
Analysis of miRNA gene targets. It was found that among all the gene targets, 27 of them were specifically associated with AD. For dementia, 13 gene targets were identified, and for neurodegenerative diseases in general, 27 gene targets were found (**A**). Additionally, five genes, namely QSER1 (glutamine and serine-rich 1), BACE2 (beta-secretase 2), RNF157 (ring finger protein 157), PTMA (prothymosin alpha), and GJD3 (gap junction protein delta 3) were identified as common target genes for the miRNAs (miRNAs that exclusively interact with circPSEN1s indicated with a red rectangle) analyzed. These genes were targeted by multiple miRNAs (six), indicating their potential importance in the regulatory network mediated by these miRNAs (**B**).

**Figure 3 biomolecules-13-01401-f003:**
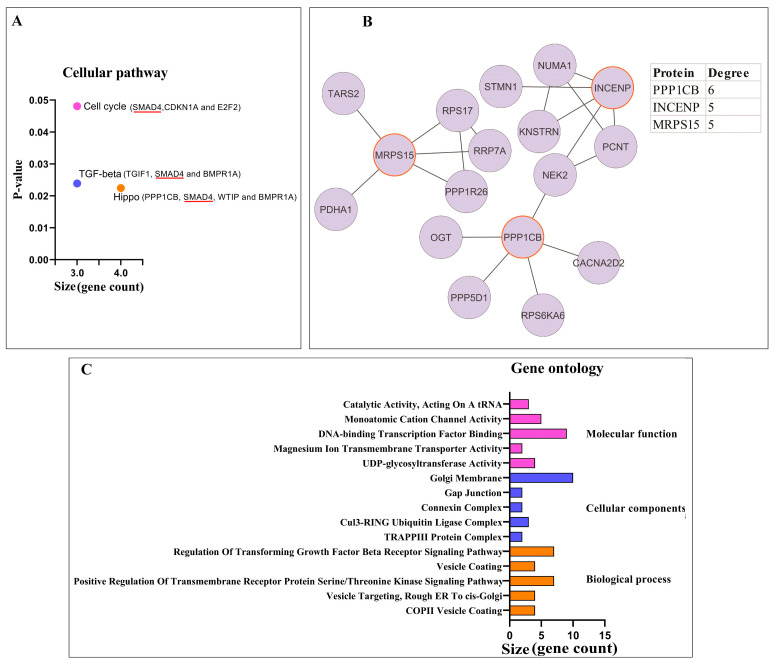
Among the miRNA gene targets, three cellular pathways showed significant enrichment (*p* < 0.05). These pathways are the cell cycle, Hippo, and TGF-β (transforming growth factor β) signaling pathways. The size axis indicates the number of genes significantly associated with each pathway. As depicted, SMAD4 (SMAD family member 4) is the only gene present in all three signaling pathways (**A**). The protein–protein interaction (PPI) network constructed for the proteins in these three signaling pathways showed that PPP1CB (protein phosphatase 1 catalytic subunit beta), INCENP (inner centromere protein), and MRPS15 (mitochondrial ribosomal protein S15) exhibited the highest degree, indicating a greater number of interactions with other proteins in the network (**B**). Bar charts visualize the distribution of enriched GO (Gene Ontology) terms and identify clusters of related genes (**C**). This suggests that the miRNAs targeted by circPSEN1s regulate several cellular pathways, potentially influencing various biological processes and signaling cascades.

**Figure 4 biomolecules-13-01401-f004:**
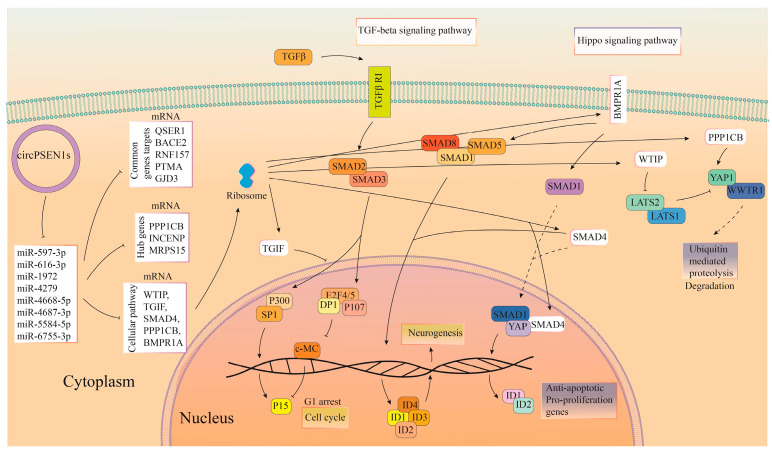
The predicted function of miRNAs interacting with circPSEN1s is summarized in this picture. circPSEN1s act as sponges for specific miRNAs, including miR-597-3p, miR-616-3p, miR-1972, miR-4279, miR-4668-5p, miR-4687-3p, miR-5584-5p, and miR-6755-3p. This interaction leads to the rescue of their target mRNAs, which can be either specific (WTIP (WT1 interacting protein), TGIF (TGFB induced factor homeobox 1), SMAD4 (SMAD family member 4), PPP1CB (protein phosphatase 1 catalytic subunit beta), and BMPR1A (bone morphogenetic protein receptor type 1A)) or common (QSER1 (glutamine and serine-rich 1), BACE2 (beta-secretase 2), RNF157 (ring finger protein 157), PTMA (prothymosin alpha), and GJD3 (gap junction protein delta 3)). As a result, the TGF-beta (transforming growth factor β) and Hippo (hippopotamus) signaling pathways are predominantly influenced.

**Figure 5 biomolecules-13-01401-f005:**
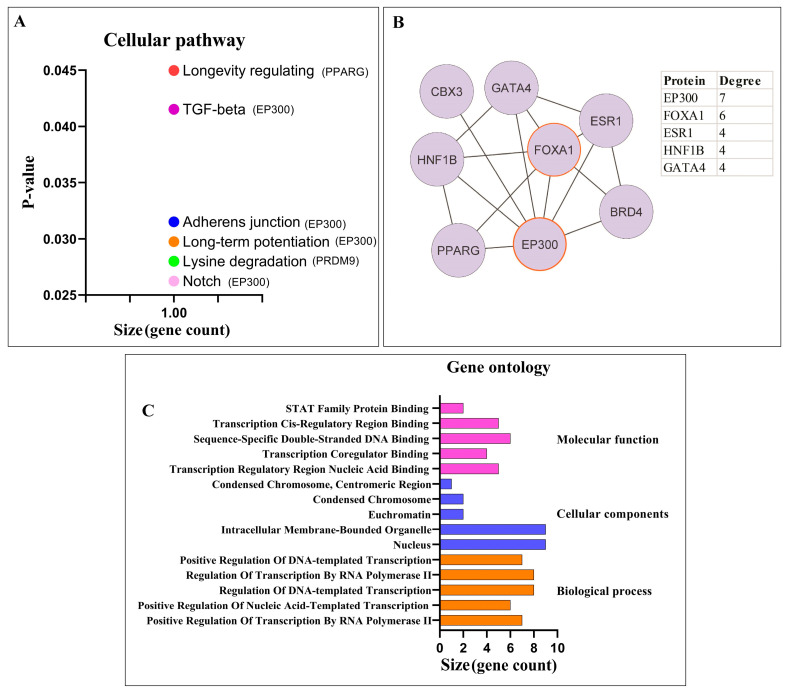
Cellular pathways and PPI for proteins that interact directly with circPSEN1s. (**A**) Analysis of proteins interacting with circPSEN1s revealed six significant cellular pathways (*p* < 0.05). These pathways are adherens junction, long-term potentiation, longevity regulating, lysine degradation, Notch signaling, and TGF-β (transforming growth factor β) signaling. The size axis indicates the number of genes significantly associated with each pathway. As depicted, EP300 (E1A binding protein p300) is the only gene present in four signaling pathways. (**B**) The protein–protein interaction (PPI) network constructed for the proteins interacting with circPSEN1s showed that FOXA1 (forkhead box A1) and EP300 exhibited the highest degree, indicating a greater interaction with other proteins in the network. The high degree of interaction suggests that FOXA1 and EP300 may play important roles in mediating the functions of circPSEN1s and their associated pathways. (**C**) Gene ontology analysis was performed to investigate the functional characteristics of proteins interacting with circPSEN1s.

**Figure 6 biomolecules-13-01401-f006:**
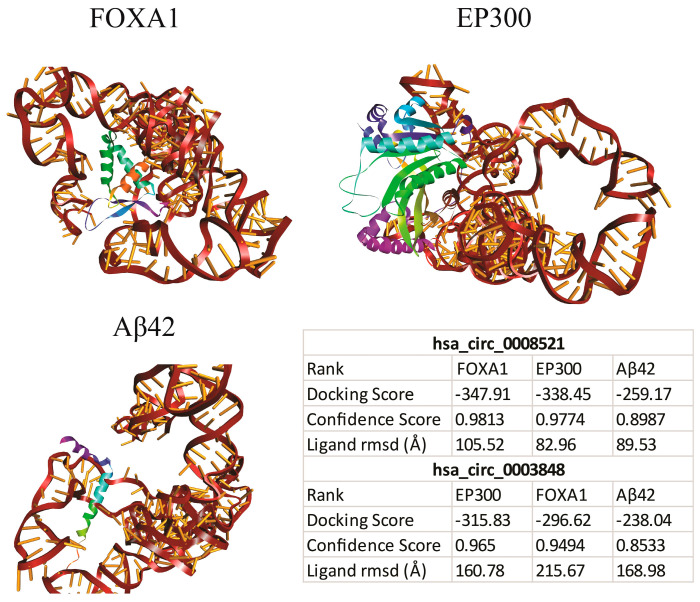
Interaction of circPSEN1s with EP300 (E1A binding protein p300), FOXA1 (forkhead box A1), and Aβ42 and docking scores. These scores indicate the binding affinity of circPSEN1s with the respective proteins. Based on the docking scores, hsa_circ_0008521 shows more affinity (higher negative docking scores) toward FOXA1 than EP300 and Aβ42, and hsa_circ_0003448 shows more affinity (higher negative docking scores) toward EP300 than FOXA1 and Aβ42.

**Figure 7 biomolecules-13-01401-f007:**
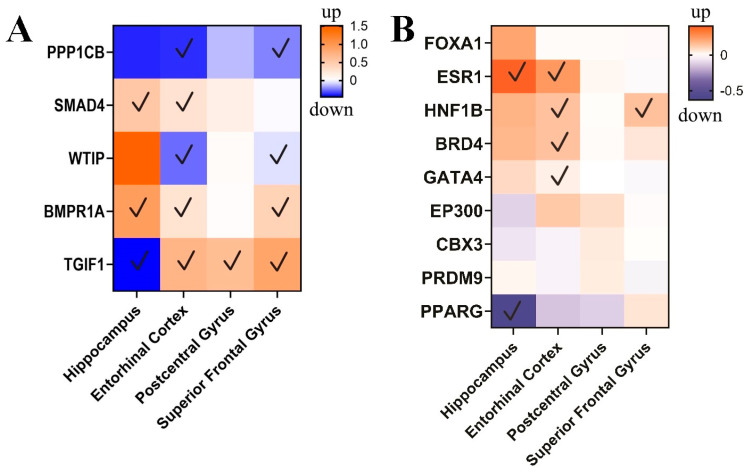
The expression levels of genes associated with circPSEN1 (PPP1CB (protein phosphatase 1 catalytic subunit beta), SMAD4 (SMAD family member 4), WTIP (WT1 interacting protein), BMPR1A (bone morphogenetic protein receptor type 1A), and TGIF1 (TGFB-induced factor homeobox 1)) were analyzed, and their differential expression patterns were visualized in a heatmap. The heatmap represents changes in gene expression in AD compared to control samples. Among the four brain regions analyzed, the entorhinal cortex exhibited the highest number of significant changes (indicated with a checkmark) in the expression of circPSEN1-related genes. Notably, TGIF1 showed significant expression changes in all four brain regions, suggesting its potential role in AD pathogenesis across different brain areas (**A**). The gene expression of proteins that directly interact with circPSEN1s is depicted here, showing a higher expression of the targeted proteins in the entorhinal cortex of the brain (**B**).

**Figure 8 biomolecules-13-01401-f008:**
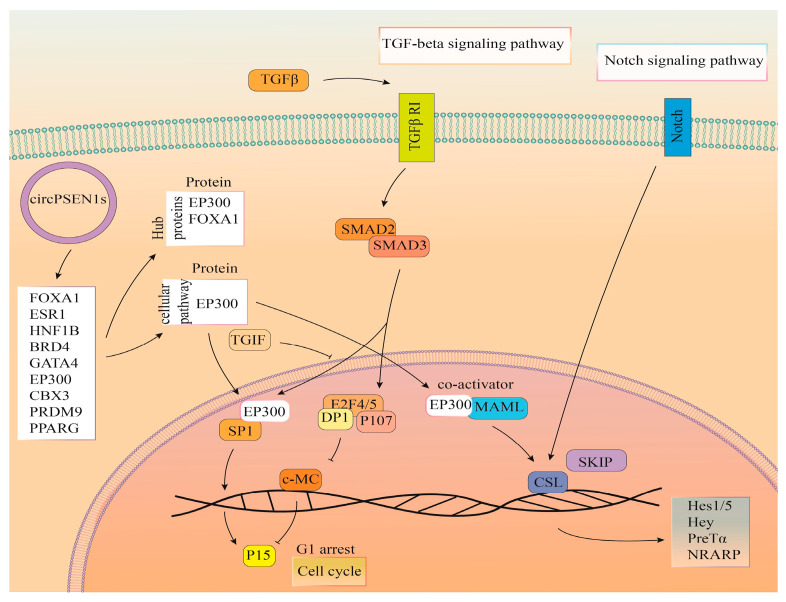
This picture summarizes the predicted function of proteins interacting with circPSEN1s. It illustrates that circPSEN1s directly interact with specific proteins, namely FOXA1 (forkhead box A1), ESR1 (estrogen receptor 1), HNF1B (hepatocyte nuclear factor-1 beta), BRD4 (bromodomain containing 4), GATA4 (GATA binding protein 4), EP300 (E1A binding protein p300), CBX3 (chromobox 3), PRDM9 (PR/SET domain 9), and PPARG (peroxisome proliferator-activated receptor gamma). Consequently, these interactions predominantly influence the TGF-beta and Notch signaling pathways. Considering the notably elevated *p*-value associated with the Notch pathway and the well-established significance of TGF-β as a pathway indirectly regulated by circPSEN1s’ targeted miRNAs, we opted to give prominence to the depiction of these two pathways in the image. This decision is aimed at ensuring clarity and focus, avoiding unnecessary complexity.

## Data Availability

Data included in the article/referenced in the report.

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
