# Peer review of "Presenilin-1-Derived Circular RNAs: Neglected Epigenetic Regulators with Various Functions in Alzheimer’s Disease"

_biomolecules, 2023, doi:10.3390/biom13091401_

Round 1
Reviewer 1 Report
In the present manuscript, Sanadgol and colleagues studied PSEN1 gene-derived circular RNAs (circPSEN1s) and their potential role in Alzheimer's disease (AD). In particular, they focused on two circPSEN1 species (hsa_circ_0008521 and hsa_circ_0003848), identified also by Chen et al., 2022 https://doi.org/10.1186/s40478-022-01328-5. The novelty of this work is the detailed bioinformatic analysis the authors performed to predict circPSEN1s-miRNAs and circRNA-protein interaction as well as circPSEN1s-proteins molecular docking. Additionally, they evaluated miRNAs gene-target prediction, pathway enrichment and microarray data of postmortem AD patients.
The manuscript is well written and the experimental design is clearly reported. I recommend it for publication.
Author Response
I appreciate your thoughtfulness and insightful feedback.
Reviewer 2 Report
This in silico study aimed to elucidate the possible molecular mechanism of two circPSEN1 RNAs found to be upregulated in AD. The extensive study is well structured, follow a clear path and utilize a large number of data-bases, online platforms. The results are clearly presented (e.g. nice, understandable figures). My major concern is that two miRNA were identified as specific targets for circ- versus “normal” PSEN1 RNA, but all 8 non-specific miRNA were studied further. Please, clarify! The discussion is presently a list of random information, it should be compiled. I suggest to focus more on direct and miRNA-transmitted protein interactions, the similarities and differences, try to give more comprehensive explanation and utility of the findings.
Specific suggestions:
No abbreviation list is added to each figure legends.
Line 25 “PPI network exhibited EP300 and FOXA1 have the highest number of protein interactions.”?
Line 27 confirmed
Line 34 “mutually TGF-β” ?
Line 45-49: Can be deleted as the important of AD was made clear before.
Line 57 to the cerebellum
Line 61 “PSEN1 is involved in the C-terminal transmembrane region” part of the region? Associated to this region? Actually this sentence can be omitted as previously it was clarified that PSEN1 is important to APPA cleavage.
Line 63 If circRNA might contains exon, why are they non-coding? Can this bond never open and result in protein synthesis?
Line 76 I would add “with presumably important role in AD.”
Line 107 add meaning of KEGG (Kyoto Encyclopedia of Genes and Genomes, it should be KEGG-based pathways…
Line 148. Were these hub proteins chosen based upon previous e.g. PPI analysis?
Line 150 Was the aim of this study to confirm previous, in silico data on human samples? I suggest starting with the sentence line 162-164 and clarify this question.
Line 169. Is this location/length different from the non-circular PSEN1?
Line 175. AS Aβ is expressed also at the periphery, it would be interesting to see, whether these circRNAs are exclusive to brain. Are they typical to neurones (or can be found in glial cells)?
Line 177 I think it would be important to note that these data based upon the consensus of three databases (if I understood correctly).
Line 190 Why not only the 2 miRNAs, specific for circRNAs were studied further? I think the other 6 is not relevant here… Or it should be added, which gene targets were uniqu to these two miRNAs.
Line 224 Were these the two specific miRNAs?
Line 229 This statement is true if they are associated with the two specific miRNA.
Line 250 GO terms? Clarify the abbreviation! (gene ontology)
Line 254 The cell cycle as third component on Fig 3A is not mentioned here.
Line 263 If I understood correctly so far indirect protein interaction through miRNAs were studied and here the direct protein interaction of circRNAs were evaluated. Please, add this clarification.
Line 264 Similarly the found interactions for each model can be described to miRNAs as well in line 177.
Line 270-274 Useless sentences, can be deleted.
Which proteins interacted also with the gene's mRNA?
Line 277 Add: directly interacting proteins. Emphasize the similarities and difference between the direct and indirect protein networks.
Line 279 “l7sine degradation” ?
Line 280 What does this preference means? (see e.g. number of connection on fig 3B).
Line 285 abbreviations? (e.g. Forkhead box protein A1 (FOXA1))
Line 289 Delete, does not belong to results. Instead write something about the functions, and their relations to previously mentioned proteins.
Fig 5A What is the x axes (all 1.00 value)? What is the meaning of greater values on Fig 3a?r values
What are in brackets? (an example protein of the group?) Wha only EP300 is there (repeatedly), but not FOXA1?
Line 309 Why Abeta (and not e.g. APP)? What is the additional value of this analysis to previously showed cricPSEN1- hub protein interactions?
Line 311-315 The numbers are given on Figure 6, useless to describe them here.
Line 315 The negative docking scores indicate stronger interaction, right? Were these differences significant? (e.g. fold change?)
Line 320 What is Picture 4? Refer to Figure 8 (however it should be figure 7, and appear earlier, closer to first mentioning). Why only TNFbeta and Notch signalization are mentioned as on Fig 5 A more pathways are presented. Moreover, it is not described in the text what could be the relation between these pathways and the proteins. It would be more interesting to summarize Fig 4 and 8 and see the direct and indirect protein interactions and there possible overlapping.
Line 332 Add rationale of this study (to confirm in silico data on human samples?). It was not clear from the method, how these microarray results were related to circPSEN1? (in the results the main focus on control-AD difference, as far as I understood, only these differences are presented on Fig 7 as well). Also in Line 340 no mentioning any relation to circPSEN1. If there is no relation, better delete this whole section (or try to connect the observed changes to expression changes related to PSEN1 before).
Line 362 First 3 sentences should be deleted. Start with the description of the results (is it starting in line 365?)
Line 372 Before writing about miRNA interacting proteins, discuss interacting miRNA. What is known about these 8-2 miRNA? What can be the difference between the 6 interacting also with the “normal” PSEN1 and the 2 exclusively acting to circPSEN1?
Line 374, 381 etc. give the meaning of abbreviation at first appearance, not here.
Line 375: It would be nice to see a Table with all these proteins, add there and not in the text the chromosome and major functions. Thereby the text can be significantly shortened.
Line 384, 387 use introduced abbreviation of Aβ
Line 389 Give possible rationale what could be the meaning of circPSEN1 -BACE1 interaction (prevent/promote activity, exacerbate/prevent A-beta accumulation).
Line 389 These general, meaningless phrases should be avoided throughout the ms.
Line 404 “cellular processes” is also too general.
Line 409 Do not introduce again already used abbreviation.
Line 412 instead of this useless sentence the authors can speculate that “as AD is the type 3 diabetes mellitus, it is clear that problems with energy supply is a key step in the development of the disease. Therefore, PTMA with mitochondrial influence can also have a role in disease development (protecting or aggravating role?).
Line 419-422 delete!
Line 423 Conclusion of proteins are missing, why cell cycle is started to be discuss here (not too much is written in the results).
Line 432 YAP and TAZ are mentioned here first. Were they not studied (although they are on Figure 4), in case so why? (similarly MST1/2 and LATS1/2 etc.)
Line 442 I would put it before line 428
Line 448- again mention direct-indirect interactions.
Line 451 delete sentence
Minor editing of English language required
Author Response
This in silico study aimed to elucidate the possible molecular mechanism of two circPSEN1 RNAs found to be upregulated in AD. The extensive study is well structured, follow a clear path and utilize a large number of data-bases, online platforms. The results are clearly presented (e.g. nice, understandable figures). My major concern is that two miRNA were identified as specific targets for circ- versus “normal” PSEN1 RNA, but all 8 non-specific miRNA were studied further. Please, clarify! The discussion is presently a list of random information, it should be compiled. I suggest to focus more on direct and miRNA-transmitted protein interactions, the similarities and differences, try to give more comprehensive explanation and utility of the findings.
Thank you for your thoughtful review and helpful insights. We have incorporated a new part into Figure 1 regarding, two miRNAs exclusively interacting with circPSEN1s and elaborated on its details within the manuscript. Additionally, our focus has shifted towards exploring direct and miRNA-mediated protein interactions. We have highlighted the commonalities and distinctions between these interactions, aiming to provide a more comprehensive understanding and practical implications of our findings.
Specific suggestions:
No abbreviation list is added to each figure legends.
We added all abbreviations to each figure legend.
Line 25 “PPI network exhibited EP300 and FOXA1 have the highest number of protein interactions.”?
We rewrite these results.
Line 27 confirmed
We corrected this word.
Line 34 “mutually TGF-β” ?
We corrected this word.
Line 45-49: Can be deleted as the important of AD was made clear before.
We removed this sentence.
Line 57 to the cerebellum
We corrected this word.
Line 61 “PSEN1 is involved in the C-terminal transmembrane region” part of the region? Associated to this region? Actually this sentence can be omitted as previously it was clarified that PSEN1 is important to APPA cleavage.
We removed this sentence.
Line 63 If circRNA might contains exon, why are they non-coding? Can this bond never open and result in protein synthesis?
We answered your question.
Line 76 I would add “with presumably important role in AD.”
We added these words.
Line 107 add meaning of KEGG (Kyoto Encyclopedia of Genes and Genomes, it should be KEGG-based pathways…
We added an abbreviation.
Line 148. Were these hub proteins chosen based upon previous e.g. PPI analysis?
Yes, we describe it.
Line 150 Was the aim of this study to confirm previous, in silico data on human samples? I suggest starting with the sentence line 162-164 and clarify this question.
We describe this concern there.
Line 169. Is this location/length different from the non-circular PSEN1?
Actually not, We answered your question in the paper.
Line 175. AS Aβ is expressed also at the periphery, it would be interesting to see, whether these circRNAs are exclusive to brain. Are they typical to neurones (or can be found in glial cells)?
Line 177 I think it would be important to note that these data based upon the consensus of three databases (if I understood correctly).
We mentioned this concern in the paper.
Line 190 Why not only the 2 miRNAs, specific for circRNAs were studied further? I think the other 6 is not relevant here… Or it should be added, which gene targets were uniqu to these two miRNAs.
We describe this data more through the manuscript.
Line 224 Were these the two specific miRNAs?
We describe it.
Line 229 This statement is true if they are associated with the two specific miRNA.
We describe it.
Line 250 GO terms? Clarify the abbreviation! (gene ontology)
We added an abbreviation.
Line 254 The cell cycle as third component on Fig 3A is not mentioned here.
We added it.
Line 263 If I understood correctly so far indirect protein interaction through miRNAs were studied and here the direct protein interaction of circRNAs were evaluated. Please, add this clarification.
We added this clarification.
Line 264 Similarly the found interactions for each model can be described to miRNAs as well in line 177.
We added this clarification.
Line 270-274 Useless sentences, can be deleted.
We deleted these sentences.
Line 277 Add: directly interacting proteins. Emphasize the similarities and difference between the direct and indirect protein networks.
We added this clarification.
Line 279 “l7sine degradation” ?
We corrected it.
Line 280 What does this preference means? (see e.g. number of connection on fig 3B).
We added this clarification.
Line 285 abbreviations? (e.g. Forkhead box protein A1 (FOXA1))
We added an abbreviation.
Line 289 Delete, does not belong to results. Instead write something about the functions, and their relations to previously mentioned proteins.
We deleted these sentences.
Fig 5A What is the x axes (all 1.00 value)? What is the meaning of greater values on Fig 3a?r values. What are in brackets? (an example protein of the group?) Wha only EP300 is there (repeatedly), but not FOXA1?
The "size" typically refers to the number of genes or proteins found in a specific pathway and is significantly associated with each pathway indicated in brackets. As depicted, EP300 (E1A binding protein p300) is the only gene present in four signaling pathways.
Line 309 Why Abeta (and not e.g. APP)? What is the additional value of this analysis to previously showed cricPSEN1- hub protein interactions?
Considering that APP is a transmembrane protein located within the cellular plasma membrane, assessing its interaction with circular RNA is not logical. However, Aβ peptides have the ability to aggregate, leading to the formation of insoluble amyloid plaques. These Aβ peptides can also readily interact with other molecules, such as circular RNAs. We added this clarification in the manuscript.
Line 311-315 The numbers are given on Figure 6, useless to describe them here.
We removed the numbers.
Line 315 The negative docking scores indicate stronger interaction, right? Were these differences significant? (e.g. fold change?)
Indeed, you're correct. It appears that we cannot detect significant changes in these interactions.
Line 320 What is Picture 4? Refer to Figure 8 (however it should be figure 7, and appear earlier, closer to first mentioning). Why only TNFbeta and Notch signalization are mentioned as on Fig 5 A more pathways are presented. Moreover, it is not described in the text what could be the relation between these pathways and the proteins. It would be more interesting to summarize Fig 4 and 8 and see the direct and indirect protein interactions and there possible overlapping.
We added this clarification in the manuscript.
Line 332 Add rationale of this study (to confirm in silico data on human samples?). It was not clear from the method, how these microarray results were related to circPSEN1? (in the results the main focus on control-AD difference, as far as I understood, only these differences are presented on Fig 7 as well). Also in Line 340 no mentioning any relation to circPSEN1. If there is no relation, better delete this whole section (or try to connect the observed changes to expression changes related to PSEN1 before).
We added this clarification in the manuscript.
Line 362 First 3 sentences should be deleted. Start with the description of the results (is it starting in line 365?)
We removed the sentences.
Line 372 Before writing about miRNA interacting proteins, discuss interacting miRNA. What is known about these 8-2 miRNA? What can be the difference between the 6 interacting also with the “normal” PSEN1 and the 2 exclusively acting to circPSEN1?
We added this clarification in the manuscript.
Line 374, 381 etc. give the meaning of abbreviation at first appearance, not here.
We corrected it accordingly.
Line 375: It would be nice to see a Table with all these proteins, add there and not in the text the chromosome and major functions. Thereby the text can be significantly shortened.
You are right but we need to completely rewrite the manuscript if want to perform this modification.
Line 384, 387 use introduced abbreviation of Aβ
We corrected it accordingly.
Line 389 Give possible rationale what could be the meaning of circPSEN1 -BACE1 interaction (prevent/promote activity, exacerbate/prevent A-beta accumulation).
We corrected it accordingly.
Line 389 These general, meaningless phrases should be avoided throughout the ms.
We corrected ms accordingly.
Line 404 “cellular processes” is also too general.
We removed the sentence.
Line 409 Do not introduce again already used abbreviation.
We corrected it accordingly.
Line 412 instead of this useless sentence the authors can speculate that “as AD is the type 3 diabetes mellitus, it is clear that problems with energy supply is a key step in the development of the disease. Therefore, PTMA with mitochondrial influence can also have a role in disease development (protecting or aggravating role?).
We corrected it accordingly.
Line 419-422 delete!
We removed the sentence.
Line 423 Conclusion of proteins are missing, why cell cycle is started to be discuss here (not too much is written in the results).
We corrected it accordingly.
Line 432 YAP and TAZ are mentioned here first. Were they not studied (although they are on Figure 4), in case so why? (similarly MST1/2 and LATS1/2 etc.)
We removed the sentence.
Line 442 I would put it before line 428
We corrected it accordingly.
Line 448- again mention direct-indirect interactions.
We corrected it accordingly.
Line 451 delete sentence
We removed the sentence.
Reviewer 3 Report
This manuscript presented the findings from in-silico analysis of cirPSEN1. It only included human microarray data of less than 100 patients.
I have the following concern on the methodology and analysis:
1. there are many post-mortem microarray data available at GEO, the authors are expected to include more data for analysis
2. The details of setting of the tool of analysis were not clear.
It is not clear the finding is well linked with human data
Author Response
This manuscript presented the findings from in-silico analysis of cirPSEN1. It only included human microarray data of less than 100 patients.
Our approach did not involve comparing various microarray datasets. Instead, we selected the most comprehensive and complete dataset to validate our in silico analysis. Moreover, the selected dataset comprises 253 samples from four different brain areas.
I have the following concern on the methodology and analysis:
- there are many post-mortem microarray data available at GEO, the authors are expected to include more data for analysis
Our approach did not involve comparing various microarray datasets.
- The details of setting of the tool of analysis were not clear.
We added this information to part 2.5. of the manuscript.
It is not clear the finding is well linked with human data
We corrected the discussion accordingly and added related data.